evolution/bioinformatics/statistics

amino acids, codon, protein sequence, evolution, Poisson distribution, conditions

**Author for correspondence:**
Guoqiu Wu
e-mail: wgq@seu.edu.cn

# The constraints between amino acids influence the unequal distribution of codons and protein sequence evolution

Yi Qian[1], Rui Zhang[2], Xinglu Jiang[2] and Guoqiu Wu[3,4]

[1]Department of General Surgery, Zhongda Hospital, [2]Medical School, [3]Center of Clinical Laboratory Medicine, Zhongda Hospital, Medical School of Southeast University, and [4]Jiangsu Provincial Key Laboratory of Critical Care Medicine, Southeast University, 87 Ding Jiaqiao, Nanjing 210009, People's Republic of China

GW, 0000-0002-8859-8654

Four nucleotides (A, U, C and G) constitute 64 codons at free combination but 64 codons are unequally assigned to 21 items (20 amino acids plus one stop). About 500 amino acids are known but only 20 are selected to make up the proteins. However, the relationships between amino acid and codon and between 20 amino acids have been unclear. In this paper, we studied the relationships between 20 amino acids in 33 species and found there were three constraints between 20 amino acids, such as the relatively stable mean carbon and hydrogen (C : H) ratios (0.50), similarity interactions between the constituent ratios of amino acids, and the frequency of amino acids according with Poisson distribution under certain conditions. We demonstrated that the unequal distribution of 64 codons and the choice of amino acids in molecular evolution would be constrained to remain stable C : H ratios. The constituent ratios and frequency of 20 amino acids in a species or a protein are two determinants of protein sequence evolution, so this finding showed the constraints between 20 amino acids played an important role in protein sequence evolution.

## 1. Introduction

The rates and sequences of 20 amino acids in proteins have remained a central subject in evolutionary and molecular biology for half a century [1–4]. By far, protein expression level and the functional importance of a protein have been viewed to two major determinants [5]. About 500 amino acids are known but only 20 are selected to make up the proteins, an important sort of biological polymers [6]. Darwin's theory of evolution and

neutral theory, two major rival theories, can be used to explain some laws of species evolution or have been justified by some natural phenomena [7–9]. The rate of amino acid reflects both Darwinian selection for functionally advantageous mutations and selectively neutral evolution operating within the constraints of structure and function [10]. However, relationships and interactions between amino acids in molecular evolution are scarcely reported.

Codons, three nucleotides, locate in transfer RNA (tRNA) molecules to carry amino acids and to read the mRNA at a time. Four nucleotides (A, U, C and G) constitute 64 codons at free combination. However, 64 codons are unequally assigned to 21 items (20 amino acids plus one stop). The explanations about this phenomenon are unclear, such as the frozen accident hypothesis [11], stereochemical hypothesis [12], coevolution hypothesis [13], ATP-centric hypothesis [14] and so on. Based on these observations, we should found a point to establish relationship between 20 amino acids and their corresponding codons and the carbon and hydrogen (C : H) ratios were proved to be a good choice by repeated simulation calculation in this paper.

The molecular evolutionary clock is that the rate of evolution at the molecular level is approximately constant through time and among species [15]. Biologists can compare protein sequences, such as haemoglobins, cytochrome c and fibrinopeptides from different species of mammals, to infer the dates of major species divergence events in the Tree of Life [16–18]. However, multiple factors were found to influence the varying molecular evolutionary rates among species, which could lead the clock to be violated, including generation time, population size, basal metabolic rate and so on [15,19,20]. Next-generation sequencing technologies have led to the increased availability of genomic data offering molecular clock dating studies and some effective methods also have been reported, such as relaxed clock models, but still the divergences do not disappear [21,22]. To find the relationship between 20 amino acids or approximately constant of molecular evolution under what conditions could be helpful to better application of molecular clock.

To seek the similarities of organisms on DNA and protein level should be a method to research on their relationships. Life on Earth probably began 3.5–4 billion years ago [23]. Some similarities have been retained for such long time of evolution, and it definitely needs constraint forces [5,24–26]. We collected protein sequences from 33 species of genome data from the National Center for Biotechnology Information (NCBI) (see electronic supplementary material, table S1) and made some statistical analysis to seek these similarities. In this paper, we reported three similarities of 33 species in molecular evolution, which was defined as the constraints between 20 amino acids in protein evolution.

## 2. Results and discussion

The amino acid gain and loss in protein evolution were reported to be a universal trend via comparing 12 available triplets of complete prokaryotic genomes and not to be driven by any simple trend at the DNA level [27]. Here, we set the relationships between the sum of 20 amino acids of all proteins in a species and the number of their corresponding codons to explore the amino acid gain and loss (see Methods). Under ideal condition, there was a linear relationship between the two after random mutation, so we made the rate of their corresponding codons as a reference standard to compare the amino acid gain and loss. In figure 1a, we found four amino acids, such as Met (M, CG% = 33.3%), Asp (D, CG% = 50%), Glu (E, CG% = 50%) and Phe (F, CG% = 16.7%), in all 33 species were on the red line ($y = 0$), which showed that they are 'gainers', while seven amino acids, such as Trp (W, CG% = 66.7%), Cys (C, CG% = 50%), His (H, CG% = 50%), Thr (T, CG% = 50%), Pro (P, CG% = 83.3%), Ser (S, CG% = 50%) and Arg (R, CG% = 72.2%), in all 33 species were under the red line ($y = 0$), which showed that they are 'loser'. The biggest variation is Lys (K, from −0.181 to 1.68, CG% = 16.7%) and the smallest variation is Thr (T, from −0.413 to −0.181, CG% = 50%). GC% of four gainers are less than or equal to 50% and GC% of seven losers are greater than or equal to 50%, which showed protein evolution was driven by any simple trend at the DNA level and more amino acids encoded by (G + C)-rich codons would lose.

Carbon, hydrogen, nitrogen, oxygen and sulfur composition of 20 amino acids constitute the variety of proteins. The carbon and hydrogen (C : H) ratios are from 0.40 (Gly, 2/5) to 0.92 (Trp, 11/12) and the mean C : H ratio of 20 amino acids is 0.54, while it is 0.50 (theoretical value) if the constituent ratio of 20 amino acids happens to be the rates of their corresponding codons in a protein. Then, we respectively calculated the sum of 20 amino acids of all proteins in each species and calculated the mean C : H ratios of each species. However, regardless of the significant change rates of 20 amino acids in different species (figure 1a), the carbon and hydrogen ratios kept on a relatively stable value (0.50 ±

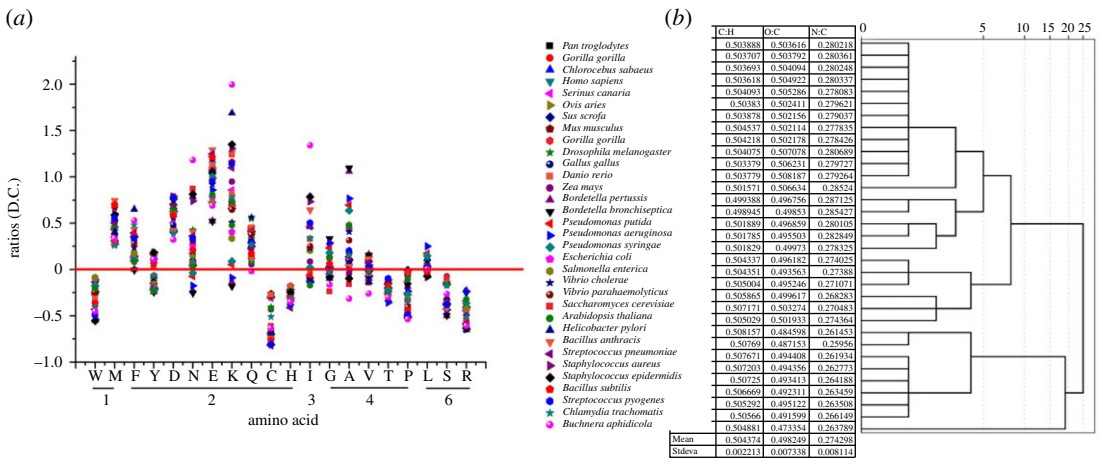

**Figure 1.** The mean of carbon and hydrogen ratios keep a relatively stable value (0.50) in 33 species regardless of the significant change rates of 20 amino acids. (a) Relationships between the sum of 20 amino acids of all proteins in 33 species and the number of their corresponding codons ($F = (n * 61/N * c) - 1$), N: the sum of 20 amino acid in the genome of a species; n: the number of one amino acid in the genome of a species; c: the number of their corresponding codons; 61 is the sum codons. (b) the mean of carbon and hydrogen ratios (0.50), oxygen and carbon ratios (0.50), nitrogen and carbon ratios (0.27) in 33 species.

0.002, figure 1b). It was miraculous that although loss and gain of amino acids resulted in a significant change of their constituent ratios, C : H ratios of 33 species remained at the theoretical value (0.50). Here, we suspected carbon and hydrogen ratios were a constraint between 20 amino acids in protein evolution and a determinant of the rate of protein sequence evolution. Furthermore, the relationships and interactions between amino acids in molecular evolution would be well explained based on the C : H ratios. These constraints between amino acids would be conducive to understand that 64 codons are unequally assigned to 20 amino acids, 20 amino acids selected to the composition of proteins and the protein sequence evolution.

To further validate the authenticity of this conclusion, we obtained the bivariate correlations of 20 amino acids in 33 species shown in figure 2a. We found Trp (W, C : H 0.917, the highest C : H ratio) was in negative correlation to Phe (F, C : H 0.818, the second-highest C : H ratio) and Tyr (Y, C : H 0.818, the second-highest C : H ratio) and positive correlation to Gly (G, C : H 0.4, the lowest C : H ratio, $r = -0.705$, $p = 0.000$), Ala (A, C : H 0.429, $r = -0.641$, $p = 0.000$) and Arg (R, C : H 0.429, $r = -0.843$, $p = 0.000$). All amino acids with C : H ratios more than 0.5 were found to be in a positive correlation with at least one amino acid with C : H ratios less than 0.50 (F and K; Y and N; H and C; D and V; E and Q; P and W; figure 2a). In other words, the accumulation or reduction of an amino acid in a species can influence others in order to keep the balance of C : H ratios. We set the relationships between the C : H ratios and the amount of codon (figure 2b), O : C ratios (electronic supplementary material, figure S1a) and N : C ratios (electronic supplementary material, figure S1b) of 20 amino acids to explore whether the distribution of 64 codons in 20 amino acids concerned the balance of C : H ratios. Twenty points in figure 2b are under the red line unlike electronic supplementary material, figure S1a and S1b. There was a negative correlation between the amount of codon and C : H ratios ($r = -0.504$, $p = 0.023$). The relationships between C : H ratios and O : C ratios (figure 2c) and between C : H ratios and N : C ratios (figure 2a) were similar to those between the amount of codon and C : H ratios, so we found that O : C ratios and N : C ratios also remained stable and were 0.50 and 0.27, respectively (figure 1b). The distribution of 64 codons and the choice of 20 amino acids in molecular evolution would be constrained to remain stable C : H, O : C and N : C ratios.

*Homo sapiens* have an estimated 20 000–25 000 genes [28], and in this paper, we have collected 21 560 protein sequence from NBCI. The C : H ratios of 21 560 proteins were calculated (see Method) and their distribution diagram (mean: 0.50, 95% confidence interval: 0.5040 to 0.5044) is shown in figure 3a. A leptokurtosis appeared above the normal distribution curve (red line, skewness = 0.103 ± 0.017, kurtosis = 2.498 ± 0.033), which illustrated the C : H ratios of proteins tend to be 0.50. Skewness and kurtosis of 33 species are shown in figure 3b and 3c. The absolute value of their skewness were less than 0.6 while kurtosis of 15 species was greater than 1 and of all species was greater than 0.25. When the absolute value of both skewness and kurtosis is less than 1, the data fit normal distribution (17 species).

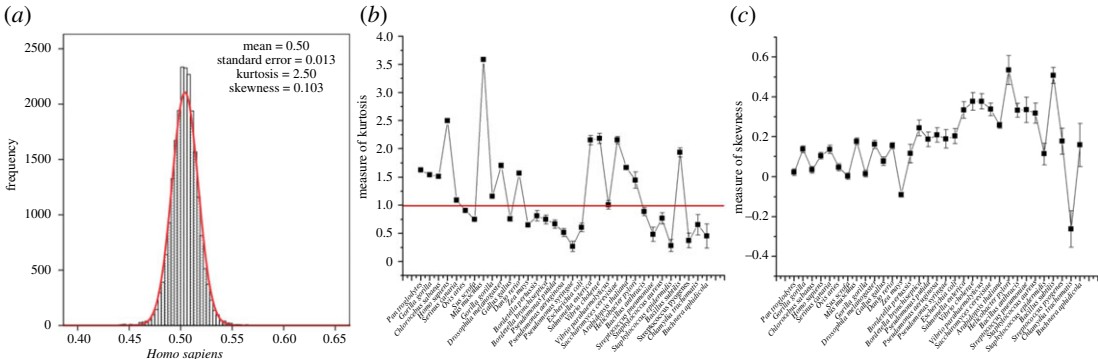

**Figure 2.** The reasons of the relatively stable mean of carbon and hydrogen ratios. Relationships between 20 amino acids in 33 species (*a*), between the amount of codons and C : H ratio in 20 amino acids (*b*), between O : C and C : H ratio in 20 amino acids (*c*) and between N : C and C : H ratio in 20 amino acids (*d*).

**Figure 3.** The carbon and hydrogen ratios of all proteins in 33 species were tend to 0.50. (*a*) The histogram of C : H ratio of 21 560 proteins in *Homo sapiens* (red line: normal curve); (*b*) the kurtosis (red line: $y = 1$) of the histogram of C : H ratio of 33 species d; (*c*) the skewness of the histogram of C : H ratio of 33 species.

The expression level and functional importance of proteins were reported to be a major determinant of the rate of protein sequence evolution [5]. However, little attention was given to the relationship between 20 amino acids that are the composition of thousands and thousands of proteins. Based on the above findings, 12 540 Pearson correlations ($20 \times 19 \times 33$) between the constituent ratios of 20 amino acids of all proteins respectively in 33 species were obtained and drawn into a heat map (figure 4*e*). Four common relationships in *Homo sapiens* are shown in figure 5*a,b*, such as a positive correlation ($r > 0.3$, $p < 0.001$; figure 5*a*) and negative correlation ($r < -0.3$, $p < 0.001$; figure 5*b*), weak correlation ($|r| < 0.3$, $p < 0.001$; figure 5*d*) and indifference correlation ($p > 0.001$; figure 5*c*). We found there were similar relationships between amino acids in all 33 species (yellow or blue lines in

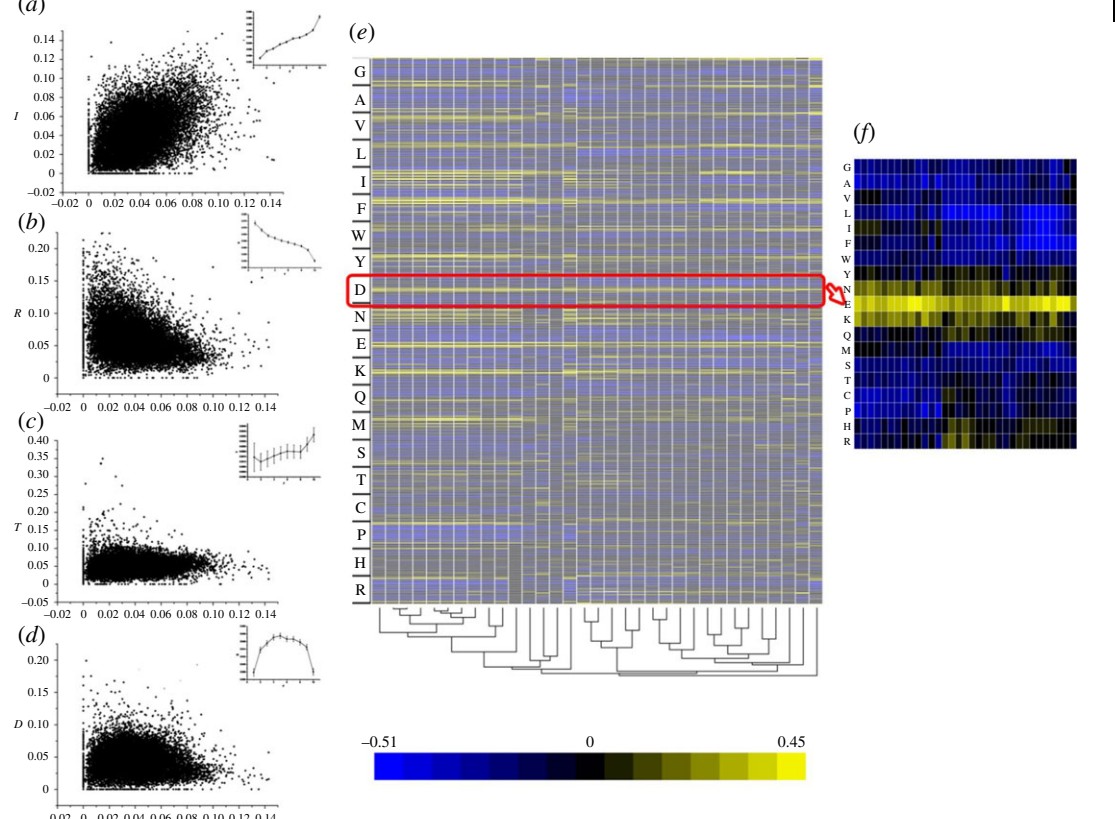

**Figure 4.** The interrelations between 20 amino acids in 33 species. (*a–d*) four kinds of scatter diagrams in *Homo sapiens* (positive correlation: $r > 0.3$, $p < 0.001$; negative correlation: $r < -0.3$, $p < 0.001$; weak correlation: $|r| < 0.3$, $p < 0.001$ and indifference correlation: $p > 0.001$); (*e*) Pearson correlation analysis between 20 amino acids in 33 species and 12 540 correlation coefficient ($r$, $19 \times 20 \times 33$) were obtained and shown in the matrix, and a phylogenetic tree was below; (*f*) one of the matrix (*e*): Pearson correlation analysis between Asp(D) and other 19 amino acids and the matrix of 627 correlation coefficient.

figure 4*e*), especially, in closely related ones. In figure 4*f*, there were relationships between Asp (D) and other 19 amino acids, and we found that Glu was in positive correlation to it in all 33 species (yellow squares in figure 4*f*). The neutral theory asserts that the vast majority of intraspecific polymorphisms and interspecific differences in protein sequence are selectively neutral rather than adaptive [29,30], which conflicts with our findings. In fact, the ratios of amino acids were correlated to others in both the species and protein level. It was a pity that we had not found the direct evidence to support that their interactions in the latter were also related to keep the balance of the C : H ratios.

For example, the number of Gly (G) in the proteins of human was $31.98 \pm 37.11$ (from 0 to 1483), and its histogram is shown in electronic supplementary material, figure S2*a*. The length of proteins in a human was $477.3 \pm 481.8$ and its histogram is shown in electronic supplementary material, figure S2*b*. Kolmogorov–Smirnov (KS) test was performed to verify whether those data fitted normal distribution or Poisson distribution. We found the two sets of data did not match the above distribution (all $Z \geq 27.84$, $P < 0.000$). We set a model that the proteins were supposed to 20 difference figures which were corresponding to the number of 20 amino acids in this protein (see figure 5*a* and Method), and we counted the frequency of the figures (0 to 35) in every protein of 33 species and the mean frequency of 0 to 35 in 33 species were shown in figure 5*b*. The most frequent is 4, 5 and 6 (vertex coordinate) and their mean value is from 0.62 (*Serinus canaria*) to 1.25 (*Bacillus anthracis*). On the left of vertex coordinate, when the figure is more than 10, their mean values become similar or equivalent in 33 species. When the actual mean values are equal to the theoretical value ($\lambda_{mean} = \lambda$), the data fit Poisson distribution (a random event), which was shown in figure 5*c*. In figure 5*d*, the frequencies of figures (6 and 20) in human are shown, and we found that the mean of the figure (6) repeated two, three and four times in a protein increased while the mean of the figure (20) repeated two, three and four times was similar with the theoretical value (Poisson distribution).

To find and explore the similarities of species in evolution may be a way of solving the mystery of the origin of organisms. In the twentieth century, the continued perfection of genetic central dogma is an

R. Soc. Open Sci. 8: 201852

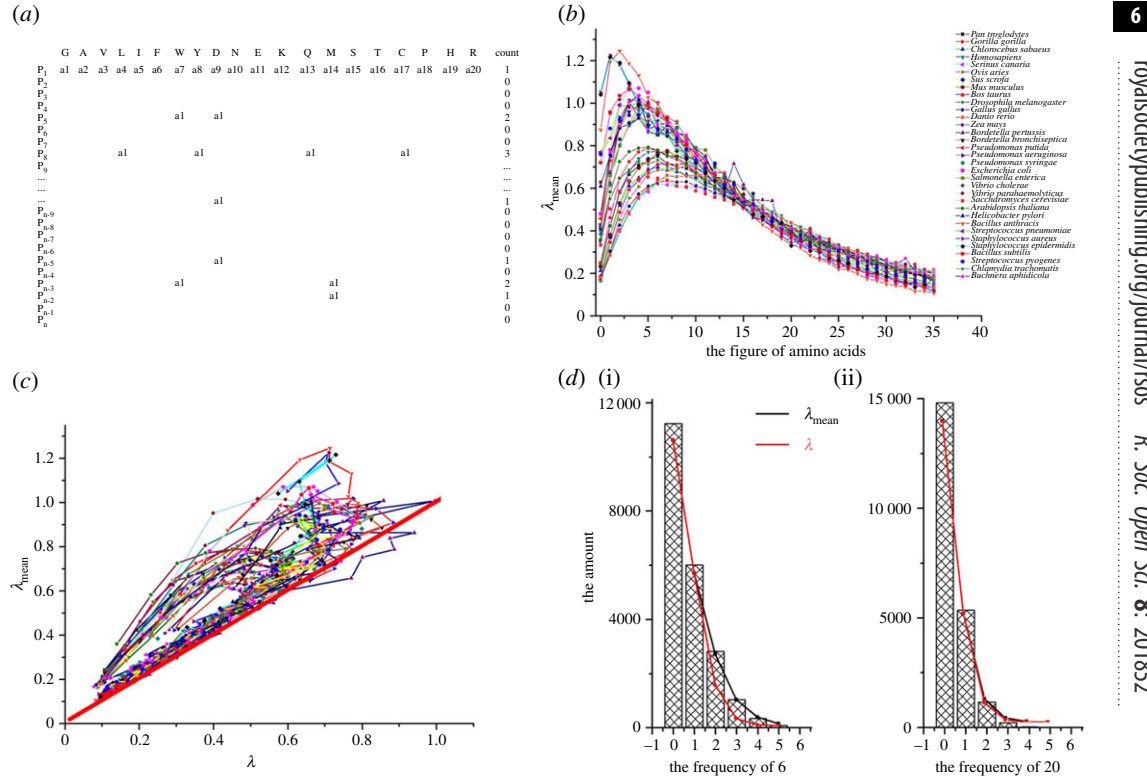

**Figure 5.** Poisson distribution. (a) The model to count the frequency of the figures which is the number of 20 amino acids in a protein; (b) the mean frequency of the figure ($\lambda_{mean}$) in 33 species, and the highest $\lambda_{mean}$ was 4, 5 or 6 in 33 species; (c) the relationship between $\lambda_{mean}$ and the theoretical frequency of the figures ($\lambda$) when these figures would fit Poisson distribution, red line: $y = x$; (d) the histogram of the frequency of 6 and 20 and their fitted curve under the conditions of $\lambda_{mean}$ and $\lambda$.

example. It is very difficult to find the similarities of species in the base and protein sequences because of biodiversity. In the past 15 years, the increased availability of genomic data for species and biological information technology offer favourable conditions. In this paper, we study the interrelation between 20 amino acids through the collection of genomic data of 33 different species. Three similarities of species as a result of constraints between amino acids in evolution were found, such as relatively stable carbon and hydrogen ratios, their interactions and Poisson distribution. These constraints are conducive to understand 64 codons are unequally assigned to 20 amino acids, 20 amino acids selected to the composition of proteins, and the protein sequence evolution.

# 3. Methods

All 33 species of genomes were extracted from the NCBI genome database (http://www.ncbi.nlm.nih. gov/genome/), such as *Pan troglodytes* (1), *Gorilla gorilla* (2), *Chlorocebus sabaeus* (3), *Homo sapiens* (4), *Serinus canaria* (5), *Ovis aries* (6), *Sus scrofa* (7), *Mus musculus* (8), *Gorilla gorilla* (9), *Drosophila melanogaster* (10), *Gallus gallus* (11), *Danio rerio* (12), *Zea mays* (13), *Bordetella pertussis* (14), *Bordetella bronchiseptica* (15), *Pseudomonas putida* (16), *Pseudomonas aeruginosa* (17), *Pseudomonas syringae* (18), *Escherichia coli* (19), *Salmonella enterica* (20), *Vibrio cholerae* (21), *Vibrio parahaemolyticus* (22), *Saccharomyces cerevisiae* (23), *Arabidopsis thaliana* (24), *Helicobacter pylori* (25), *Bacillus anthracis* (26), *Streptococcus pneumoniae* (27), *Staphylococcus aureus* (28), *Staphylococcus epidermidis* (29), *Bacillus subtilis* (30), *Streptococcus pyogenes* (31), *Chlamydia trachomatis* (32) and *Buchnera aphidicola* (33). Then we removed repeating gene sequences and collected the Protein product (NCBI Reference Sequence: NP_ or XP_).

The sequence of proteins in 33 species of genomes was collected via NCBI Reference Sequence from the NCBI protein database (http://www.ncbi.nlm.nih.gov/protein/). The number of 20 amino acids in every protein was calculated and analysed by statistics (Dryad, Dataset, https://doi.org/10.5061/dryad. xpnvx0kf8). According to neutral theory, mutations as kind of a random event, the rates of 20 amino acid

in a species should be similar to the rates of their corresponding codons, and their difference was denoted by

$$F = \frac{n * 61}{N * c} - 1,$$

where $N$ is the sum of 20 amino acids in the genome of a species; $n$ is the number of one amino acid in the genome of a species; and c is the number of their corresponding codons; 61 is the sum of codons (excluding three termination codons).

The number of carbon was by

$$C = 2 \times G + 3 \times A + 5 \times V + 6 \times L + 6 \times I + 9 \times F + 11 \times W + 9 \times Y + 4 \times D + 4 \times N + 5 \times E + 6 \times K$$
$$+ 5 \times Q + 5 \times M + 3 \times S + 4 \times T + 3 \times C + 5 \times P + 6 \times H + 6 \times R$$

where the abbreviation of 20 amino acid is their number in a protein, such as $G : C_2H_5NO_2$, $A : C_3H_7NO_2$, $V : C_5H_{11}NO_2$, $L : C_6H_{13}NO_2$, $I : C_6H_{13}NO_2$, $F : C_9H_{11}NO_2$, $W : C_{11}H_{12}N_2O_2$, $Y : C_9H_{11}NO_3$, $D : C_4H_7NO_4$, $N : C_4H_8N_2O_3$, $E : C_5H_9NO_4$, $K : C_6H_{14}N_2O_2$, $Q : C_5H_{10}N_2O_3$, $M : C_5H_{11}O_2NS$, $S : C_3H_7NO_3$, $T : C_4H_9NO_3$, $C : C_3H_7NO_2S$, $P : C_5H_9NO_2$, $H : C_6H_9N_3O_2$, $R : C_6H_{14}N_4O_2$. The number of hydrogen, oxygen and nitrogen were counted using the same method.

In this paper, we set a model that every protein was supposed to be made from 20 figures which is the number of 20 amino acids (figure 5a). The figures are from 0 to 33. The frequency of those figures appearing in a protein was calculated in 33 species to find their similarities under what conditions they accord with the Poisson distribution.

Poisson distribution is a discrete probability distribution that expresses the probability of a given number of events occurring in a fixed interval of time and/or space if these events occur with a known average rate and independently of the time since the last event. The probability of observing $k$ events in an interval is given by the equations

$$P(X = k) = \frac{\lambda^k}{k!}e^{-\lambda},$$

$$P(X = k) = \frac{\lambda}{k}P(X = k - 1)$$

and

$$P(X = 0) + P(X = 1) + P(X = 2) + P(X = 3) + \cdots + P(X = k) = 1,$$

where $\lambda$ is the average number of events per interval; $e$ is the number 2.71828 …(Euler's number) the base of the natural logs; $k$ takes values 0, 1, 2, … ; and $k!$ is the factorial of $k = k \times (k - 1) \times (k - 2) \times \cdots \times 2 \times 1$.

When $\lambda_{mean} = \lambda = (P(X = k)/P(X = k - 1))k$, it accords with the Poisson distribution.

$$\lambda_{mean} = P(X = 0) \times 0 + P(X = 1) \times 1 + P(X = 2) \times 2 + P(X = 3) \times 3 + \cdots + P(X = k) \times k.$$

In this paper, all statistical analysis was performed with SPSS 19.0. The bivariate correlation was analysed via the Pearson correlation coefficient and normal and Poisson distribution test were used KS test.

Data accessibility. The number of 20 amino acids in every protein of 33 species of genomes (Dryad, Dataset, https://doi.org/10.5061/dryad.xpnvx0kf8) [31].

Authors' contributions. Y.Q. carried out data collection, participated in data analysis and drafted the manuscript; R.Z. carried out the statistical analyses and drafted the manuscript; X.J. collected field data and revised the manuscript; G.W. designed the study and critically revised the manuscript.

Competing interests. We declare we have no competing interests.

Funding. This work was supported by the National Natural Science Foundation of China (grant nos. 81603016 and 81773624), the Natural Science Foundation of Jiangsu Province (grant nos. BK20160706 and BE2017746) and the National Science and Technology Major Project (grant nos. 2018ZX09301026-005 and 2020ZX09201015).

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
