## [Peer Review File · Royal Society Open Science]

Review History

RSOS-201852.R0 (Original submission)

Review form: Reviewer 1

Is the manuscript scientifically sound in its present form?

Yes

Are the interpretations and conclusions justified by the results?

Yes

Is the language acceptable?

Yes

Do you have any ethical concerns with this paper?

No

Have you any concerns about statistical analyses in this paper?

No

Recommendation?

Accept as is

Comments to the Author(s)

The authors proposed an interesting hypothesis in this manuscript. They showed that a stable C/H ratio plays an important role during the molecular evolution. Although more sample number are preferred to have a better clue, this manuscript can inspire more people working on this topic. I recommend accepting this manuscript.

Review form: Reviewer 2

Is the manuscript scientifically sound in its present form?

No

Are the interpretations and conclusions justified by the results?

No

Is the language acceptable?

Yes

Do you have any ethical concerns with this paper?

No

Have you any concerns about statistical analyses in this paper?

No

Recommendation?

Major revision is needed (please make suggestions in comments)

Comments to the Author(s)

In this paper, professor Guoqiu Wu and colleague pointed out that the usage of 20 amino acids in the proteome of 33 organisms deviates from the theoretical value calculated from the number of codons. In addition, the authors found that the C/H ratio of proteome of 33 organisms were surprisingly highly conserved. The authors investigated the relationship between the usage of 20 amino acids and the C/H ratio. They argued that the usage of amino acids whose C/H ratio are greater than 0.5 were positively correlated with those whose C/H ratio are less than 0.5. They argued that this positive correlation suggests that there is a constraint that balances the C/H ratio to be constant. The authors also analyzed the distribution of amino acid frequency usage in all human proteins.

Comments:

In this paper, the authors have shown the data and calculations; however, they have not write about the reason why the C/H ratio may work as a constraint. If the authors discuss the reason, it may encourage further studies.

In my interpretation, some data and explanations seem to be inconsistent. I recommend the authors to rewrite these points.

On line 52 of page 4, the author wrote "which showed protein evolution was driven by any simple trend at the DNA level and more amino acids encoded by (G + C)-rich codons." However,

the four amino acids called "gainers" had less than 50% GC content. Therefore, "(A + U) -rich codons" can be correct.

On page 5, line 30, the author wrote, "All amino acids with C: H ratios more than 0.5 were found to be positive correlation with at least one amino acid with C: H ratios less than 0.50", and gave us the examples, "F and M, Y and M, H and V, D and K, E and R, P and R; see Figure 2a". However, in Figure 2a, the pairs H and V, D and K, E and R show a negative correlation. Moreover, in this explanation, the authors picked up some examples from the data (Figure 2a) and used them as evidence. However, counterexamples were also found from the data (Figure 2a). In detail, F and R, Y and R, P and I (F, Y, and P are amino acids with a CH ratio greater than 0.5, and R and I are amino acids with a CH ratio less than 0.5.) show a negative correlation. It is not good to cut out only a part of the data and use it as evidence. It is necessary for the authors to analyze the whole data and make discussion.

Decision letter (RSOS-201852.R0)

Dear Dr Qian

The Editors assigned to your paper RSOS-201852 "The constraints between amino acids influence the unequal distribution of codons and protein sequence evolution" have now received comments from reviewers and would like you to revise the paper in accordance with the reviewer comments and any comments from the Editors. Please note this decision does not guarantee eventual acceptance.

Both reviewers are positive about your findings and the manuscript. However, one reviewer raises a number of points that you should address in your revision. We invite you to respond to the comments supplied below and revise your manuscript. Below the referees' and Editors' comments (where applicable) we provide additional requirements. Final acceptance of your manuscript is dependent on these requirements being met. We provide guidance below to help you prepare your revision.

Please submit your revised manuscript and required files (see below) no later than 21 days from today's (ie 09-Feb-2021) date. Note: the ScholarOne system will 'lock' if submission of the revision is attempted 21 or more days after the deadline. If you do not think you will be able to meet this deadline please contact the editorial office immediately.

on behalf of Steve Brown (Subject Editor)
openscience@royalsociety.org

Reviewer comments to Author:

Reviewer: 1

Comments to the Author(s)

The authors proposed an interesting hypothesis in this manuscript. They showed that a stable C/H ratio plays an important role during the molecular evolution. Although more sample number are preferred to have a better clue, this manuscript can inspire more people working on this topic. I recommend accepting this manuscript.

Reviewer: 2

Comments to the Author(s)

In this paper, professor Guoqiu Wu and colleague pointed out that the usage of 20 amino acids in the proteome of 33 organisms deviates from the theoretical value calculated from the number of codons. In addition, the authors found that the C/H ratio of proteome of 33 organisms were surprisingly highly conserved. The authors investigated the relationship between the usage of 20 amino acids and the C/H ratio. They argued that the usage of amino acids whose C/H ratio are greater than 0.5 were positively correlated with those whose C/H ratio are less than 0.5. They argued that this positive correlation suggests that there is a constraint that balances the C/H ratio to be constant. The authors also analyzed the distribution of amino acid frequency usage in all human proteins.

Comments:

In this paper, the authors have shown the data and calculations; however, they have not write about the reason why the C/H ratio may work as a constraint. If the authors discuss the reason, it may encourage further studies.

In my interpretation, some data and explanations seem to be inconsistent. I recommend the authors to rewrite these points.

On line 52 of page 4, the author wrote "which showed protein evolution was driven by any simple trend at the DNA level and more amino acids encoded by (G + C)-rich codons." However, the four amino acids called "gainers" had less than 50% GC content. Therefore, "(A + U) -rich codons" can be correct.

On page 5, line 30, the author wrote, "All amino acids with C: H ratios more than 0.5 were found to be positive correlation with at least one amino acid with C: H ratios less than 0.50", and gave us the examples, "F and M, Y and M, H and V, D and K, E and R, P and R; see Figure 2a". However, in Figure 2a, the pairs H and V, D and K, E and R show a negative correlation.

Moreover, in this explanation, the authors picked up some examples from the data (Figure 2a) and used them as evidence. However, counterexamples were also found from the data (Figure

2a). In detail, F and R, Y and R, P and I (F, Y, and P are amino acids with a CH ratio greater than 0.5, and R and I are amino acids with a CH ratio less than 0.5.) show a negative correlation. It is not good to cut out only a part of the data and use it as evidence. It is necessary for the authors to analyze the whole data and make discussion.

===PREPARING YOUR MANUSCRIPT===

===PREPARING YOUR REVISION IN SCHOLARONE===

<https://royalsociety.org/journals/authors/author-guidelines/#supplementary-material> to include a suitable title and informative caption. An example of appropriate titling and captioning may be found at https://figshare.com/articles/Table_S2_from_Is_there_a_trade-off_between_peak_performance_and_performance_breadth_across_temperatures_for_aerobic_sc_ope_in_teleost_fishes_/3843624.

Author's Response to Decision Letter for (RSOS-201852.R0)

See Appendix A.

RSOS-201852.R1 (Revision)

Review form: Reviewer 2

Is the manuscript scientifically sound in its present form?

Yes

Are the interpretations and conclusions justified by the results?

Yes

Is the language acceptable?

Yes

Do you have any ethical concerns with this paper?

No

Have you any concerns about statistical analyses in this paper?

No

Recommendation?

Accept as is

Comments to the Author(s)

In the first review, I wrote three major comments to the authors. Now I have confirmed that authors rewrote the sentences or replied to my comments about these three points.

Decision letter (RSOS-201852.R1)

Dear Dr Qian,

It is a pleasure to accept your manuscript entitled "The constraints between amino acids influence the unequal distribution of codons and protein sequence evolution" in its current form for publication in Royal Society Open Science. The comments of the reviewer(s) who reviewed your manuscript are included at the foot of this letter.

Please ensure that you send to the editorial office an editable version of your accepted manuscript, and individual files for each figure and table included in your manuscript. You can send these in a zip folder if more convenient. Failure to provide these files may delay the processing of your proof.

Please see the Royal Society Publishing guidance on how you may share your accepted author manuscript at <https://royalsociety.org/journals/ethics-policies/media-embargo/>. After publication, some additional ways to effectively promote your article can also be found here

<https://royalsociety.org/blog/2020/07/promoting-your-latest-paper-and-tracking-your-results/>.

on behalf of Professor Steve Brown (Subject Editor)
openscience@royalsociety.org

Reviewer comments to Author:

Reviewer: 2
Comments to the Author(s)

In the first review, I wrote three major comments to the authors. Now I have confirmed that authors rewrote the sentences or replied to my comments about these three points.

Appendix A

Dear Editors and Reviewers:

Thank you for your letter and comments concerning our paper RSOS-201852 "The constraints between amino acids influence the unequal distribution of codons and protein sequence evolution". We have revised the manuscript according to your kind advices and reviewers' detailed suggestions. Now, we resubmit the revised manuscript to RSOS. The main corrections in the paper were used highlighting and the responds to the reviewer's comments are as flowing in a point-by-point manner.

Thank you for your consideration. I look forward to hearing from you.

Sincerely,

Guoqiu Wu, Professor

a. Medical School of Southeast University, Nanjing 210009, People's Republic of China.

b. Center of Clinical Laboratory Medicine, Zhongda Hospital, Southeast University, Nanjing 210009, People's Republic of China.

Tel: 86-025-83272150

E-mail: nationball@163.com

Responds to the reviewer's comments:

Reviewer #1:

The authors proposed an interesting hypothesis in this manuscript. They

showed that a stable C H ratio plays an important role during the molecular evolution. Although more sample number are preferred to have a better clue, this manuscript can inspire more people working on this topic. I recommend accepting this manuscript.

Response: Thanks for the reviewer's suggestion and recommendation.

Reviewer #2:

Comment 1. In this paper, the authors have shown the data and calculations; however, they have not write about the reason why the C/H ratio may work as a constraint. If the authors discuss the reason, it may encourage further studies.

Response: Thanks for the reviewer's suggestion. In the results and discussion, we have neglected to discuss the reason why the C/H ratio may work as a constraint. Hence, we are sorry about our mistake. The reason and background of this study have been described in the introduction. In this paper to find a point to establish relationship between 20 amino acids and their corresponding codons, the carbon and hydrogen (C:H) ratios were proved to be a good choice by repeated simulation calculation. If we would find the relationship between 20 amino acids or approximately constant of molecular evolution under what conditions, it can be helpful to better application of molecular clock. Life on earth probably began 3.5-4 billion years ago. Some similarities have been retained for such long time of evolution and it definitely needs constraint forces. We collected protein

sequences from 33 species of genome data from NCBI and made some statistical analysis to seek these similarities. In this paper, we reported three similarities of 33 species in molecular evolution, which was defined as the constraints between 20 amino acids in protein evolution. In the revised manuscript, the content of the discussion about the reason were introduced.

Comment 2. In my interpretation, some data and explanations seem to be inconsistent. I recommend the authors to rewrite these points.

Response: Thanks for the reviewer's suggestion. About these problems, we ask all authors to read the article again and find the inconsistent points. After discussion, we rewrite these points. We apologize for mistakes resulting from our carelessness.

Comment 3. On line 52 of page 4, the author wrote "which showed protein evolution was driven by any simple trend at the DNA level and more amino acids encoded by (G + C)-rich codons." However, the four amino acids called "gainers" had less than 50% GC content. Therefore, "(A + U) -rich codons" can be correct.

Response: Thanks for the reviewer's suggestion. In this sentence, some words have been accidentally deleted. "more amino acids encoded by (G + C)-rich codons" change to "more amino acids encoded by (G + C)-rich codons would lose"

Comment 4. On page 5, line 30, the author wrote, "All amino acids with C: H ratios more than 0.5 were found to be positive correlation with at least

one amino acid with C: H ratios less than 0.50", and gave us the examples, "F and M, Y and M, H and V, D and K, E and R, P and R; see Figure 2a". However, in Figure 2a, the pairs H and V, D and K, E and R show a negative correlation.

Response: Thanks for the reviewer's suggestion. Sorry for this mistake. "F and M, Y and M, H and V, D and K, E and R, P and R" change to "F and K, Y and N, H and C, D and V, E and Q, P and W".

Comment 5. Moreover, in this explanation, the authors picked up some examples from the data (Figure 2a) and used them as evidence. However, counterexamples were also found from the data (Figure 2a). In detail, F and R, Y and R, P and I (F, Y, and P are amino acids with a CH ratio greater than 0.5, and R and I are amino acids with a CH ratio less than 0.5.) show a negative correlation. It is not good to cut out only a part of the data and use it as evidence. It is necessary for the authors to analyze the whole data and make discussion.

Response: Thanks for the reviewer's suggestion. Hereinabove, we have suspected carbon and hydrogen ratios was a constraint between 20 amino acids in protein evolution and a determinant of the rate of protein sequence evolution. In 33 species, the carbon and hydrogen ratios kept on a relatively stable value (0.50 ± 0.002), but that of 20 amino acids are 0.40 to 0.92. Hence, we divide 20 amino acids into two groups, such as the carbon and hydrogen ratio less than 0.50 and more than 0.50. Then, we have found that

amino acids with C: H ratios more than 0.5 were found to be positive correlation with at least one amino acid with C: H ratios less than 0.50. So the carbon and hydrogen ratios can maintain 50%. Due to the diversity of organism, we cannot make a clear conclusion about the relationship between amino acids.